# A New Primary Care Model Based on Population Needs: A Nationwide Cross-Sectional Study

**DOI:** 10.3390/nursrep15070250

**Published:** 2025-07-08

**Authors:** Silvia Domínguez Fernández, Pedro García Martínez, María Isabel Mármol-López, Esther Nieto García, María Begoña Sánchez Gómez, Mª Guadalupe Fontán Vinagre, Diego Ayuso-Murillo, Susana Montenegro Méndez, Francisco Javier Pérez-Rivas

**Affiliations:** 1Spanish Nursing Research Institute, General Nursing Council of Spain, 28023 Madrid, Spain; sildom01@ucm.es (S.D.F.); g.fontan@ieinstituto.es (M.G.F.V.); d.ayuso@consejogeneralenfermeria.org (D.A.-M.); 2Villa de Vallecas Municipal Center for Community Health, Madrid Salud, Madrid City Council, 28031 Madrid, Spain; 3UCM Research Group “Public Health-Lifestyles Nursing Methodology and Care in the Community Environment” Department of Nursing, Faculty of Nursing Physiotherapy and Podiatry, Complutense University of Madrid, 28040 Madrid, Spain; frjperez@ucm.es; 4Nursing School La Fe, Adscript Center University of Valencia, 46026 Valencia, Spain; garcia_pedmarb@gva.es (P.G.M.); maribelmrlp@gmail.com (M.I.M.-L.); 5Federation of Associations of Family and Community Nursing (FAECAP), 28007 Madrid, Spain; esther.nietog@gmail.com; 6Research Group GREIACC, Health Research Institute La Fe Valencia, 46026 Valencia, Spain; 7Community Nursing Association (AEC), 46014 Valencia, Spain; 8Family and Community Nurse, Lucero Health Center, Madrid Health Service (SERMAS), 28047 Madrid, Spain; 9Faculty of Health Sciences, Department of Medical Specialties and Public Health, Universidad Rey Juan Carlos (URJC), 28933 Madrid, Spain; 10Geriatric Specialist Nurse, Cieza-Este Health Center Area IX Murcia Health Service, Cieza, 30530 Murcia, Spain; mariab.sanchez15@carm.es; 11Faculty of Nursing, Catholic University of Murcia (UCAM), 30107 Murcia, Spain; 12Chair of Nursing, University of La Laguna (ULL), 38200 San Cristóbal de La Laguna, Spain; 13Research Network on Chronicity, Primary Care, and Health Promotion—RICAPPS—(RICORS), Carlos III Health Institute, 28029 Madrid, Spain; 14Institute of Health Research Hospital 12 de Octubre (Imas12), 28041 Madrid, Spain

**Keywords:** primary health care, clinical nurse specialist, community health nurse, role extension, service users’ perceptions

## Abstract

**Background/Objectives**: The role of the primary health care nurse has evolved since the Spanish Ministry of Health officially established the professional profile of the nurse specialist in Primary Health Care in 2005. Despite the potential benefits of this new professional profile in the population’s health, their actual scope of practice is still unknown and still largely underutilized. This study aimed to explore demands and expectations of adult primary health care service users regarding the role of the nurse specialist in primary health care. **Methods**: A nationwide cross-sectional study consisted of a computer-assisted telephone survey to a random sample of 1200 adults living in Spain. A self-developed 19-item instrument assessed population’s perception of the role of the nurse specialist in primary health care. Descriptive, bivariate and logistic regression models explored associations between sociodemographic characteristics with perception of the nurse specialist role. **Results**: Most participants (82.3%) would choose a nurse specialist in primary health care and consider that the nurse specialist in primary health care should expand their scope of practice requesting diagnostic test (70%) and prescribing medications for chronic diseases (63.8%). **Conclusions**: Results show a population’s positive perception towards expanding the scope of practice of the nurse specialist in primary health care in the Spanish healthcare system. Primary health care models should acknowledge the potential of expanding the competencies of this professional profile.

## 1. Introduction

In recent decades, several high-income countries have promoted the expansion of nursing competencies in primary health care (PHC), particularly in response to population aging and the increase in chronic diseases. This international trend has led to the integration of roles such as advanced practice nurses and nurse prescribers, with widely evaluated results in terms of improved access, quality of care and patient satisfaction, promoting the efficiency of the health care system. Countries such as the United States, Canada, the United Kingdom, Australia, New Zealand, Finland and the Netherlands have developed consolidated models of advanced practice nursing in Primary Health Care, enabling nurses to diagnose, prescribe and manage patients autonomously [1,2,3].

PHC in the Spanish health care system represents the basic health care services, acting as the first point of contact for people with health problems. It provides accessible, comprehensive and continuous attention throughout the patient’s life, functioning as a case manager, care coordinator, and regulator of patient flow. PHC includes activities related to health promotion, health education, disease prevention, health care delivery, health maintenance and recovery, as well as physical rehabilitation and social work [4]. All these activities are provided by interdisciplinary teams, among which 34,875 professionals belong to the nursing category [4].

In recent years, several factors such as the aging of the population, the increase in chronic diseases (since 2019, there has been a 10.84% increase in the number of people suffering from one or more chronic health conditions) and new user expectations demanding more personalized and accessible care, have exponentially increased the demand of the Spanish National Health System. This situation emphasizes the urgent need to strengthen and modernize PHC in order to adapt to new challenges and guarantee the sustainability of the health care system [5].

In Spain, the scope of practice of PHC nursing has been progressively established through subsequent regulations based on the Alma Ata Declaration [6]. Beginning with the implementation of the new PHC model launched in the 1980s, the regulation of this model by Royal Decree 137/1984, of 11 January, on basic health structures, the publication of the General Health Law [7], the incorporation of nursing studies into the university system, and the inclusion of subjects such as public health and community nursing in the curriculum, have been the factors that have contributed to laying the current foundations of family and community nursing [8].

This transition in nursing training constituted a significant shift in health care delivery, shifting its focus toward the development and evaluation of care and incorporating public health across the curriculum, contributing to improved health outcomes and promoting more efficient, evidence-based care [5].

After the Royal Decree 992/1987, of 3 July 1987 [9], approved the title of Nurse Specialist, the new Royal Decree 450/2005, of 22 April 2005 [10], on nursing specialties has finally enabled the development of the Family and Community Nursing Specialty Program. The aim of this specialized role is to provide an optimal response to the needs arising from demographic, social and economic changes that require the planning, management, and provision of effective and efficient nursing care for individuals, families, and the community.

The Royal Decree 1302/2018 [11], which regulates the nurses’ competencies in pharmacology, represents a significant step forward in allowing nurses to indicate, use, and authorize the dispensing of medicines and medical devices [12,13,14]. The development of guidelines coordinated by the Ministry of Health, enables nurses to optimize their competencies through high-value actions such as the following: educating patients in the management of pharmacological treatments, establishing strategies to improve therapeutic adherence, monitoring health outcomes, detecting adverse effects, preventing treatment-related complications, and improving accessibility by reducing unnecessary appointments or delays in initiating, modifying or discontinuing treatments [15,16,17,18,19,20,21,22,23]. In this context, and within a shared care model for addressing the health needs of the population, nurse prescribing is a fundamental tool for improving the accessibility and quality of health care. However, nursing expertise in pharmacology is not always recognized and valued by some medical groups, the media, or the public [11].

The Spanish Health Care System is divided in 17 autonomous communities, each of which has capacity to regulate health legislation. Thus, only certain autonomous communities allow nurses to prescribe medicines with the aim of improving access and the management of patients with chronic diseases. Nurse prescribing has become a well-established aspect to advance nursing clinical practice, so addressing the increasing complexity of patient care and the growing demands on PHC services [24].

Nurses Specialized in Family and Community Nursing (NSFCNs) are qualified to develop community intervention plans within the health care system and public health alert systems, as well as to conduct evaluations through epidemiological studies (looking not only disease, but also at health and care outcomes) and to assess the quality of health programs. These competencies provide a well-established evidence-based alternative to help regulate health care demand, promote self-care, reduce emergency department visits and hospital readmissions, all within a public health approach and always within their scope of practice [8,10,25].

NSFCNs are essential in transforming PHC to a person-centered model of care. Their ability to empower patients, provide comprehensive care, and lead organizational innovation positions them as key agents of change within the Spanish Health Care System [26,27].

Despite the essential role nurses play in the PHC, numerous studies have pointed to a persistent lack of professional recognition, both at the institutional level and in terms of social perception. This invisibility is reflected, for example, in the differences in prestige assigned by the general population to different health care professions, where nurses consistently score lower than doctors [28,29]. Furthermore, the representation of nurses in leadership positions or in clinical decision-making remains limited, which contributes to consolidating a subordinate position within interprofessional teams. This situation not only affects their visibility and professional status, but also has implications for their job motivation, professional development and talent retention in the health care sector [30].

This persistent undervaluation also influences public perceptions of the scope and safety of advanced nursing practices. Although social acceptance of nurses prescribing and ordering tests has increased, evidence shows that a significant proportion of patients still prefer doctors to perform these tasks, particularly in more complex clinical settings or when technical expertise is deemed crucial [31,32].

However, when it comes to patient satisfaction, particularly in PHC, a Spanish report showed that nurses scored slightly higher than doctors (8.13 vs. 8.1, respectively) [33,34]. The higher satisfaction with nurses is attributed to factors such as closer patient interaction, clearer communication, continuous presence, perceived commitment, emotional availability, and growing professional recognition [35].

Despite this, NSFCNs feel that their efforts are not adequately recognized and their participation in community health activities remains limited [36]. Most NSFCNs and residents reported a deficit in the acquisition of competencies outlined in the training program and identified irregularities in its implementation. Similarly to other health professions, NSFCNs perceive a lack of visibility of their role among the general population and public administration (64.7% of specialist nurses were working or had worked as generalist nurses in PHC, and only 14.2% were employed as specialists, while in 2014, 46% worked in PHC without a specified category) [37].

Nowadays, service users’ opinions about the nurses’ role in PHC have changed significantly, and they also perceived that the development of the nursing role has had a positive impact on their health [38]. However, a considerable lack of understanding remains regarding who nurses are, what they do, and what they are capable of achieving. This is partly due to the persistent misinformation about the actual scope of practice, which is often reduced to a narrow focus on specific tasks and activities, projecting a distorted image [39].

An informed and educated population is empowered to self-care by being able to identify and access appropriate PHC services and resources, thus preventing their underutilization [40,41,42].

Several autonomous communities are planning to grant NSFCN a greater role leading the design and implementation of community health activities, in collaboration with other team members, such as social workers. The progressive integration of NSFCNs into PHC teams offers a significant opportunity to strengthen PHC and promote nurse-led reform [5].

Nurses’ competencies can and should be used by the health care system to benefit the community, the system itself, and the nursing profession [28].

To this end, it is important to define the health care model we aim for. If we want nurses to be responsible for a number of families and a community and we implement a model with a defined reference population based on planned actions coordinated with other team professionals, we will obtain objective outcomes based on this model. However, if we adopt a task and activity-based model, the evaluation will be completely different [43].

Community health is one of the pillars for the development and strengthening of PHC, as it enables a comprehensive approach to the population’s needs, contributing to health promotion and reduce health inequalities [44]. A new model of care must be based on a strong PHC, with a public health approach, emphasizing continuity of care, quality of service and community action.

This study aims to evaluate the population’s perception of the expansion of the NSFCN’s scope of practice and, in particular, the population knowledge of the NSFCN’s competencies.

The specific aims are as follows:
-To explore adult service users’ awareness of the expanded role of the NSFCN in PHC.-To analyze the relationship between the population’s sociodemographic characteristics and their perception of NSFCN’s autonomy and competencies.-To identify the main demands of users regarding the NSFCN in PHC centers.

The underlying purpose of this study is to propose a model that enhances the contribution of NSFCN to the health system by analyzing the population’s needs and expectations regarding NSFCN’s role in PHC.

## 2. Materials and Methods

### 2.1. Study Design

This is a cross-sectional study. Data were collected through a telephone survey.

### 2.2. Questionnaire Design

The survey tool was developed through a literature review and the input of expert nurses in PHC.

The panel of expert nurses consisted of a total of 11 professionals from different Autonomous Communities, representing different professional profiles (management, research, teaching, and clinical practice). Of note, three of them were members of the Spanish General Council of Nursing, representing nurses at the national level, with the additional participation of nurses from existing family and community nursing associations in Spain: the Federation of Family and Community Nursing Associations (FAECAP) and the Community Nursing Association (AEC).

Five meetings were held with the working group to define the characteristics that the new PHC model should have. The content developed during these meetings served as the basis for drafting a set of proposed questions covering the different areas discussed.

As a method of evaluating the questions, the initial proposal was carried out individually, avoiding contact between the members of the group. The evaluation was based on content evaluation criteria, including relevance, appropriateness, clarity, correct wording and suggestions for additional questions. In the first phase, a framework was drafted by consensus based on the comments of all participants. In the second phase, efforts were focused on negotiating, developing and drafting the final questionnaire content and its various items. Once developed, the questionnaire was piloted with several patient associations to ensure that the content and language were appropriate and understandable to the general population. After a final review, the development of the questionnaire was completed, resulting in a total of 19 questions related to the socio-demographic characteristics of users and the characteristics of the nurses and the care provided.

### 2.3. Participants

Participants were selected by simple random sampling, using Random Digit Dialing (RDD) of mobile phones [45]. According to the Spanish National Institute of Statistics (INE) [46], the total resident population in Spain in April 2024 was 48,692,804, with 42,207,683 people over 18 years of age in January 2024. The calculated sample size was 385 surveys, with a confidence level of 95% and margin of error of 5%.

Inclusion criteria included being 18 years of age and older, resident in Spain, and agreeing to participate in the survey, with the assumption that all participants had potential access to a NSFCN through the public health system. Exclusion criteria included language barriers and lack of access to a mobile phone. The survey took approximately 10 to 15 min to complete.

### 2.4. Data Collection

A telephone survey conducted using a Computer-Assisted Telephone Interviewing (CATI) system allowed data collection between 7 and 18 October 2024. The company’s system generates a random database. All data were managed in accordance with Regulation (EU) 2016/679 and Organic Law 3/2018 of December 5 on the protection of personal data and the guarantee of digital rights and were not used for purposes other than this study [47,48].

This study was approved by the Research Ethics Committee of the Hospital Príncipe de Asturias in Madrid on 23 July 2024 (Approval 33/2024). Completion of the survey was considered implicit informed consent, and participants were informed of the content and confidentiality of the survey at the beginning of the telephone interview.

Phone calls and statistical analysis was carried out by Análisis e Investigación, S.L., a company that operates in accordance with the CCI/ESOMAR Code of Conduct and has an implemented Quality Management System for the design of market and public opinion studies, as well as for data collection, preparation, and processing, in accordance with the international standard ISO 20252 [49].

### 2.5. Data Analysis

Variables included socio-demographic characteristics (gender, age, Autonomous Community, level of education, type of health insurance, income level, and contact with health professionals); variables related to the frequency and type of visits to the PHC Center (frequency of visits in the past year, accompaniment during visits, and by whom); variables related to nursing care at the PHC center (knowledge of the designated nurse, frequency of nursing consultations, method of appointment request, waiting time for care, duration of consultations, alternative forms of care, types of services received and participation in group activities); variables related to the new nurse-patient care model (continuity of care by the same nurse for comprehensive care and for all family members and opinions on collaboration with expert patients to support other patients); and variables related to knowledge and acceptance of new competencies (prescriptions, diagnostic tests, post-discharge contact, care provided by nurse specialists, management of patient cases by nurses with referral to physicians when necessary, and management of PHC centers).

The statistical software Gandia BarbWin 7.0 was used for the statistical processing of numerical variables, simple and cross-tabulations of frequencies. In order to avoid selection bias, the Autonomous Communities, gender and age were reweighed to align with the distribution established by the INE, with the aim of obtaining a homogeneous and representative sample across the different regions. Descriptive statistics were carried out for all the variables.

The data were then exported to SPSS version 28.0.1.0 for further analysis. Data analysis followed a comprehensive analytical approach, including both parametric and non-parametric tests. Independent samples *t*-tests were used to detect differences in mean outcomes between two mutually exclusive groups or to detect differences between groups.

Bivariate analysis was used to assess the relationship between several dependent variables—such as knowledge of the designated nurse, participation in health education activities, preference for continuity of care with the same nurse, and perceptions of nurse specialization and management leadership—and independent socio-demographic and health system contact variables. This included age, gender, level of education, perceived financial solvency, contact with health professionals and type of health insurance. All independent variables were categorized into dichotomous groups.

Variables that showed a significant chi-square test were entered into a binary logistic regression model. For each predictor variable included in the final model, the Odds Ratio (OR) and its 95% confidence interval (CI) were calculated.

Categorical independent variables (such as age groups, educational level, or type of health insurance) were transformed into separate dichotomous dummy variables using a one-vs-rest approach. That is, each category (for example, “Age 36–45 years”) was coded as a binary variable (1 = belongs to the group, 0 = does not), and simultaneously included in the model. This strategy allows for interpreting each OR as the association between that specific category and the outcome, compared to the rest of the sample, that is, all other categories of the same variable combined, rather than to a single reference group. This approach was chosen to better capture specific group effects without implying a default reference.

## 3. Results

### 3.1. Sample Description

Of the 40,000 calls made, 1480 individuals met the inclusion criteria, of which 280 (18.9%) declined to participate. A total of 1200 individuals agreed to participate in the study. Of these, females accounted for 60.5%, 70.2% were exclusive users of the public health system, and 41.2% had professional or personal contact with health professionals. The distribution by age group and Autonomous Communities was adjusted to match the distribution provided by the INE (Table 1).

### 3.2. Visits to Primary Health Care Centers

A total of 19.1% of the surveyed population reported visiting the PHC center at least once a month, while 9.8% had not visited a PHC center in the past year. From the age of 50 onwards, the frequency of monthly visits to PHC centers increases, reaching 20% in this age group. In addition, 62.2% reported visiting PHC centers to accompany other family members, such as children or partners (Table 2).

### 3.3. Access to and Experience of the Nurse at the Primary Health Care Center

The descriptive study shows that 69.4% of the surveyed population are aware that they have a designated nurse at their PHC center. Knowledge of the designated nurse is higher among citizens aged 36 to 65 than among those under 36 and over 65 than among those under 36 and over 65. A total of 62.3% of the sample reported consulting with the nurse at least once a year. The number and frequency of visits to the nurse at PHC center increases with age. Of those who have visited the nurse at least once in their lives, 89.3% feel that the nurse dedicates an adequate amount of time to them, while 49.6% reported waiting more than two days to be seen, with a median of 2 and an interquartile range of 6 days. Appointments with the PHC nurse are primarily scheduled through three main channels: in-person at the health center, by physician referral, or online. The most common types of care provided are diagnostic and therapeutic procedures, chronic disease management, monitoring of child growth and development monitoring, and skin injury care. A total of 12.4% of the total sample reported participating in health education groups, with the most commonly attended group being related to pregnancy and childbirth care (70.5% of those attending group education sessions) (Table 3).

### 3.4. A New Model of Patient–Nurse Relationship in Primary Health Care Centers

A total of 86.6% of participants indicated that they would like very much or quite a lot to have a nurse responsible for all their personal care, and 73.3% expressed a preference for the same nurse caring for all members of their family. In addition, 66.1% of participants agreed (very much or quite a lot) with the possibility of cooperation between an expert, trained patient and a nurse (Table 4).

### 3.5. Knowledge and Acceptance of New Competences for Nurses in Primary Heatlh Care Centers

Only 4 out of 10 citizens (38.6%) are aware that nurses can prescribe certain medicines; the highest awareness of the prescribing role of nurses in PHC Centers is found among younger citizens. Most citizens believe that nurses should increase their role in the follow-up and management of chronic conditions by ordering diagnostic tests (70%) and prescribing the necessary medications for chronic diseases (63.8%). In addition, 85.3% of participants agreed that the nurse should be the first point of contact for patients discharged from hospital. Furthermore, 82.3% of respondents indicated that they would like their PHC center nurse to be a specialist (Table 5).

### 3.6. Factors Associated with the Perception and Use of Nursing Services in Primary Health Care

Analysis by gender shows that females are more aware of having a designated nurse in their PHC center (*p* < 0.01), believe they would need more consultation time (*p* < 0.01), and have participated more often in health education groups (*p* < 0.01).

Analysis by age group shows that the population aged 56–65 years (*p* = 0.02) have the greatest awareness of having a designated nurse; those aged over 46 years old are more likely to visit the nurse at least once a year (*p* < 0.01); participants aged 46–65 years are more likely to think they need more consultation time (*p* = 0.02); the group most involved in group education activities is aged 36–45 years (*p* < 0.01); and the group with the most positive perception that a nurse can manage a health center is the 18–35-year-olds (*p* < 0.01).

The population with a lower level of education is more in favor of having the same nurse care for the whole family (*p* = 0.03) and expresses more doubts about the nurse’s ability to manage a health center (*p* = 0.007) (Table 6).

### 3.7. Regression Model Results

The multiple logistic regression models included those variables that showed significant differences in the previous bivariate analyses. Results are presented as Odds Ratios (OR) with their respective 95% confidence intervals (95% CI). Although no single variable showed strong predictive power on its own, all variables were included in the models to provide a comprehensive understanding of the associations examined. Therefore, variables that did not reach statistical significance are also reported.

Categorical variables such as age group, level of education, gender, and type of health insurance were modeled using a one-vs-rest dummy coding approach. Accordingly, each odds ratio reflects the association between a specific category and all other categories of the same variable combined. This is important for interpreting the results, as no single reference group was used.

Among the most relevant findings, awareness of having a designated reference nurse consistently acts as a positive predictor across all dimensions analyzed. This awareness is significantly more frequent among individuals aged 56–65 years compared to the other age groups combined, and among those who exclusively use the public health care system compared to those who do not. Additionally, awareness is associated with greater participation in group activities, continuity of care with the same nurse, and more favorable perceptions of nurses’ leadership in primary health care center management (Table 7).

## 4. Discussion

The current Spanish PHC model is grounded in the Strategic Framework for Primary Health Care and, since its implementation, every user has been assigned both a doctor and a nurse.

This study aims to contribute to the design of a PHC model that addresses the actual needs of the population, optimizing the role of the NSFCN and promoting more accessible, comprehensive and community-based care. To achieve this, several key aspects were analyzed: the population’s knowledge of the role and function of nurses in PHC, perceptions of accessibility and quality of care, current demand for and use of nursing services, and expectations regarding a model based on nursing leadership and continuity of care. The findings provide relevant information to support the development of a model that strengthens the role of nursing in PHC, integrating both user perspectives and the available scientific evidence (Figure 1). This conceptual model highlights the key roles of the NSFCN, placing the person, family, and community at the center of care. The model integrates four main areas of action: health promotion and assets, disease prevention, self-care, and continuity of care. Each area is supported by essential components such as health education, salutogenesis, the ethical value of care, coordination of personalized care, and articulation of community resources. The outer arrows emphasize two fundamental principles of the model: accessibility and professional specialization, which ensure that nursing interventions are comprehensive and equitable. This framework aligns with population needs and the available scientific evidence, reinforcing the strategic role of nursing in PHC.

In line with this framework, the study results reveal relevant trends in users’ awareness and use of nursing services in PHC. Firstly, a significant proportion of participants (69.4%) reported being aware of having a designated nurse at their health center. In addition, more than 60% of respondents visited the nurse at least once a year, and almost a third did so two or three times a year. In terms of knowledge about nursing care according to the socio-demographic profile of the participants, it was identified that older individuals (56–65 years) and exclusive users of the public health care system are the groups who are most familiar with their designated nurse, while the 18–25 age group has the lowest level of awareness. The higher awareness among older people is probably related to their higher morbidity and prevalence of chronic conditions, which justify more frequent use of PHC services [50,51], and to the fact that the follow-up of many chronic processes is often carried out by nurses [52,53]. On the contrary, the lower awareness among younger people may be related to their limited use of PHC consultations [54], as this group generally seeks care mainly for acute conditions, which are typically managed by physicians rather than family nurses [55].

This situation could change in the coming years if the implementation of ‘nurse demand management’ processes continues, where nurses take responsibility for the management of certain minor acute conditions [56,57]. However, it is essential to recognize that the expansion of NSFCN competencies must be implemented under a paradigm that preserves the salutogenic, holistic and health-promotion approach characteristic of community nursing [8,36,58]. Nurse demand management should not translate into an excessive health surveillance model or the medicalization of natural life processes [42,57]. On the contrary, NSFCNs are especially qualified to integrate self-care, health education and community participation as central elements of their practice [40,41,56], which can contribute to de-medicalizing certain processes and empowering citizens in managing their own health [59,60,61].

The greater knowledge of the nurse’s role among those who exclusively use public health care could be explained by their socioeconomic level, as scientific literature has documented that populations with fewer resources tend to make more frequent use of PHC consultations, facilitating greater familiarity with their designated health care professionals [62,63,64].

Another relevant finding of the study is the delay in access to nursing consultations in PHC, with 49.6% of respondents reporting that they had to wait more than two days to be attended. This situation can have a negative impact on the continuity of care and health management of patients, particularly for those with chronic conditions or situations requiring early intervention. The existence of delays in nursing care highlights the need to expand and strengthen the role of the NSFCN, especially considering the anticipated assumption of new competencies and responsibilities within the health care system. An increased number of NSFCN would help to optimize consultation times, improve accessibility, and reinforce the role of nursing as a central pillar in the provision of community health care.

Regarding gender differences observed in our analysis, a particularly interesting finding is the apparent Simpson’s paradox: while bivariate analysis showed significant differences between men and women in knowledge of the designated nurse, these differences disappeared in multivariate analysis. This phenomenon suggests that gender differences are mediated by other variables such as age, educational level, or frequency of contact with health services. That is, women do not necessarily have greater knowledge about NSFCNs because they are women per se, but because they tend to use health services more frequently and act as primary family caregivers, factors that in turn are associated with greater knowledge of the health system.

In line with these findings, the predominance of women in the sample (60.5%) is a relevant methodological characteristic, as is common in health research surveys [65,66,67]. This overrepresentation could have contributed to the high level of knowledge and favorable perception of the role of NSFCNs observed, not due to gender-specific factors, but rather to the greater exposure to the health care system that characterizes this population group [51]. While this participant profile provides valuable insight into continuity of care, it also introduces a potential bias that should be considered when interpreting and generalizing the results. Future studies should seek more gender-balanced samples to confirm these findings.

In terms of the most commonly used services in nursing consultations, a significant proportion of the population seeks care for various diagnostic and therapeutic procedures, chronic disease management, child growth and development monitoring, and different types of wound care (such as wound treatment and suture management). These services have been identified in the literature as those most frequently demanded by the population [51,68]. The results also showed that greater knowledge of the family nurse’s role is associated with a more positive perception of their competencies. Participants who were aware of the nurse’s role in PHC not only expressed greater confidence in their clinical and management skills but also demonstrated greater acceptance of their autonomy in prescribing treatments and ordering diagnostic tests. This suggests that misinformation may be a barrier to the implementation of new nursing competencies. These findings have been supported by successive systematic reviews evaluating the effectiveness of nurse-led care in chronic disease management and patient satisfaction. More than 20 years ago, the review by Horrocks et al. [59] already demonstrated that nurses in PHC can provide effective care and achieve positive health outcomes comparable to those provided by physicians. Years later, Keleher et al. [69] reinforced these findings by showing that nurses are effective in care management, improving treatment adherence, and implementing prevention and health promotion strategies. Subsequently, Laurant et al. [70], in a 2018 review, confirmed that specialized nurses, such as NSFCN, provide a quality of care that is equal to or even better than that of PHC physicians, achieving similar or better health outcomes and higher levels of patient satisfaction. Finally, Lukewich et al. [60] consolidated this evidence by demonstrating that patients report a high degree of satisfaction with nurse-led care, particularly in aspects such as chronic disease management, quality of life, self-care capacity, and health behaviors.

A widely agreed aspect among participants who recognize the role of nursing in PHC is the preference for a model of care based on continuity of care. In this regard, it is highly valued that the same nurse is responsible for the patient’s comprehensive care (addressing all their health needs) and provides continuous care over time, as well as caring for all members of the family unit. This approach, which has been a fundamental pillar of PHC since its inception [71,72], is supported by scientific evidence demonstrating improved health outcomes when continuity of care is ensured. Continuity of care, understood as the provision of ongoing care by the same professional over time, enables a deeper understanding of the patient’s health conditions and their family and social context, facilitating disease prevention, early detection of health problems, reducing unnecessary medicalization, and contributing to improved quality of life and patient satisfaction [73,74]. This model is particularly beneficial for elderly individuals and patients with chronic conditions [75], who, as previously mentioned, are often cared for by nurses within the PHC setting. Similarly, it is well known that family characteristics significantly influence the biological, psychological, and social health problems of all family members. Therefore, providing care to families as a whole, as well as to each member individually, is considered fundamental, as stated in the decree SAS/1729/2010, which regulates the training program for the NSFCN [8]. Several studies recognize the essential work performed by nurses in family care, highlighting their role as key professionals in holistic health care, disease prevention, and the promotion of well-being within family and community settings [36].

The high level of acceptance of the possibility of nurses assuming leadership roles in PHC centers—particularly among those who know their designated nurse—reflects the population’s recognition that nurses with a high level of education and training are essential to the delivery of health care. It is important to remember that training in health care management is part of the nursing core curriculum, and these basic leadership and management competencies are introduced during undergraduate education [76]. This recognition is even more evident among the younger population, with the 18–35 age group showing the most positive perception. This phenomenon may be attributed to the advancement of nursing as an academic discipline, with access to master’s and doctoral studies, which has contributed to improving its professional and academic profile [77,78].

Conversely, the greater resistance to nursing leadership management observed in the older age group (30.6% opposition in ≥66 years vs. 14.5% in 18–35 years) deserves careful interpretation. This resistance could reflect traditional conceptions about professional health care hierarchies, formed during decades when nursing had a more subordinate role. However, this same group showed greater use and knowledge of the designated nurse, suggesting an ambivalent view that must be considered when designing implementation strategies adapted to different generations.

From a broader health systems perspective, the results of this study support the integration of NSFCNs in high-responsibility functions within PHC organizations. The current context of professional shortages and increasing care complexity requires efficient role redistribution and optimal utilization of the competencies of all available professional profiles. The findings align with various conceptual and strategic frameworks that emphasize the importance of multidisciplinary approaches to health care delivery.

In particular, Wagner’s Chronic Care Model (CCM) [79] and the National Health System’s Chronic Disease Management Strategy [80] highlight the need for proactive and multidisciplinary teams to improve chronic disease management. In this sense, the high citizen acceptance of continuity of care by NSFCNs (86.6%) reinforces the feasibility of one of the key pillars of these models: support for self-care sustained in lasting therapeutic relationships. Likewise, from the Quadruple Aim framework approach [81], the results suggest that the proposed model could simultaneously contribute to improving patient experience, population health outcomes, system efficiency, and professional satisfaction of health care teams.

In the context of the Spanish health system, characterized by universality, the results indicate that there is a solid social base for implementing these reforms, particularly considering that 70.2% of participants are exclusive users of the public system. Current public policies, such as the Primary and Community Care Action Plan 2025–2027 [5], already point in this direction, promoting collaborative and multidisciplinary teams where nurses play a leading role in chronic care, prevention, and community participation.

### 4.1. Implications for Clinical Practice and Health Care System Organization

Improving health care and strengthening PHC requires that NSFCN play a much more decisive role, based both on the competencies they possess and on the care needs of society. In this context, it is important to highlight that a new structure for nursing competencies should be developed, involving the progressive integration of NSFCN with specific and differentiated competences compared to non-specialist community nurses. This must be achieved through the identification and definition of job positions, adapted to the current needs of the population, ensuring that community nurses have a defined, planned, and structured role that guarantees the success of strategies developed to meet these needs [43].

To achieve this, it is necessary to regulate and reorganize the competencies of both generalists and NSFCNs within the PHC setting. It is also essential to develop new leadership roles to increase visibility, to incorporate nurses into high-responsibility positions where key decisions are made, and to enable them to participate actively, visibly, and responsibly in senior health care management on an equal footing with other professionals [82].

Based on the findings obtained, several implications for improving the PHC model have been identified. First, it is essential to increase the visibility of community nursing through an effective awareness and communication strategy, targeting different audiences and utilizing various channels, in order to raise awareness of the fundamental role of the NSFCN as a care manager and provider at the first level of care. Furthermore, consolidating the role of the NSFCN would help to meet the growing demand for more specialized and individualized care. To achieve this, it is essential to promote the regulation and differentiation of competences between generalist and specialist community nurses, ensuring an organizational structure that optimizes their integration into decision-making processes and health resource management. Finally, the development of new nursing leadership roles in PHC would contribute to increase health care system efficiency, ensuring more accessible, comprehensive, and community-based care.

### 4.2. Study Limitations

This study presents several limitations that should be taken into account. First, the cross-sectional nature of the study prevents the establishment of causal relationships between the variables analyzed. Furthermore, the use of a telephone survey may have influenced participation among certain population groups, such as those with less availability of time to respond. Although the sample size was representative, future research may need to expand the sample to explore the relationships between certain variables in more depth.

Another limitation to consider is the lack of information regarding the rural and urban distribution of the sample, which may have influenced the results due to differences in accessibility to PHC centers. This variable is relevant, as the availability and use of health services may vary significantly between rural and urban contexts, potentially affecting users’ perceptions and experiences of nursing care.

From a methodological standpoint, it is important to emphasize that the regression models presented show modest explanatory power (R^2^ between 0.043 and 0.132), which suggests that the sociodemographic variables analyzed explain only a small proportion of the variability in perceptions and knowledge about the role of NSFCNs. This result points to the influence of other factors—prior experiences, cultural beliefs, professional perception, etc.—whose exploration might require qualitative methodologies. Future research should advance toward more integrative analytical frameworks.

An important limitation of the present study is the absence of a direct comparison between citizen preferences for increasing the number of primary health care physicians and those oriented toward expanding nurses’ functions. This comparison would have al-lowed obtaining more specific evidence to guide health policy decisions and resource allocation. Future studies should incorporate this type of comparison to estimate more precisely the relative impact of different health system strengthening strategies.

## 5. Conclusions

Most service users are aware of the existence of a designated nurse at their PHC center. However, this awareness varies depending on age and the level of contact with the health care system.

The results of this study highlight the fundamental role of the NSFCN in the follow-up of chronic diseases and the implementation of diagnostic and therapeutic procedures. The high frequency of consultations among patients with chronic diseases reinforces the importance of nursing in the management of these conditions, evidencing their contribution to continuity and quality of care. These results underline the need to continue strengthening the role of nursing by promoting strategies that optimize its impact on population health management.

There is a broad consensus on the preference for the same designated nurse providing care through life and for all members of the family unit. The results reflect a high level of acceptance for nurses assuming greater leadership roles in health centers, especially among those who have regular contact with these professionals. Furthermore, the population has a positive perception of nurses’ ability to perform advanced functions, such as prescribing treatments and ordering diagnostic tests.

Overall, the results demonstrate considerable citizen acceptance toward the expanded role of NSFCNs, particularly regarding continuity of care and advanced clinical competencies. However, there are significant variations according to age and previous experience with the health system that must be considered in policy implementation. The modest explanatory capacity of the statistical models suggests that citizen perceptions about nursing are influenced by complex factors that require additional research.

It is necessary to develop comparative studies that directly evaluate citizen preferences between different primary health care strengthening strategies, including both the expansion of nursing roles and the increase in medical professionals, to adequately in-form health policy decisions and resource allocation.

In conclusion, the results demonstrate the need to consolidate a model of care in which NSFCNs play a central role, ensuring both continuity of care and a family-centered approach, while assuming new competencies and leadership positions within the health care team. To achieve this, it is necessary to implement strategies that increase the visibility of the expanded role of the NSFCN.

## Figures and Tables

**Figure 1 nursrep-15-00250-f001:**
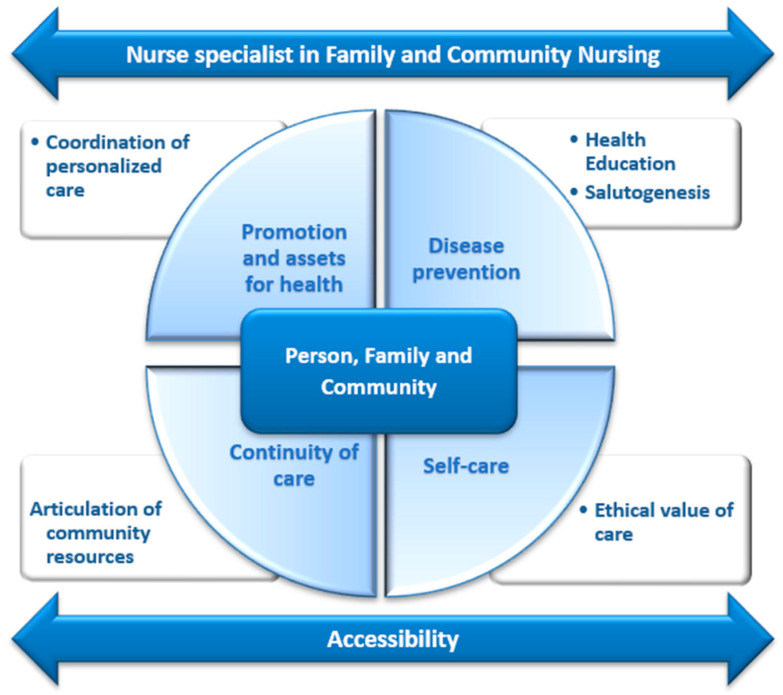
Visual representation of the proposed model.

**Table 1 nursrep-15-00250-t001:** Sociodemographic characteristics of the study population.

**IN WHICH AUTONOMOUS COMMUNITY DO YOU LIVE?**	**N**	**%**
Andalusia	212	17.7
Aragon	34	2.8
Asturias, Principality of	27	2.3
Balearic Islands	30	2.5
Canary Islands	56	4.7
Cantabria	15	1.3
Castile-Leon	62	5.2
Castile-La Mancha	51	4.3
Catalonia	195	16.3
Valencian Community	129	10.8
Extremadura	27	2.3
Galicia	71	5.9
Community of Madrid	169	14.1
Region of Murcia	37	3.1
Autonomous Community of Navarre	17	1.4
Basque Country	56	4.7
La Rioja	8	0.7
Ceuta	2	0.2
Melilla	2	0.2
**GENDER**	**N**	**%**
Female	726	60.5
Male	473	39.4
Non-binary	0	0
Prefer not to say	1	0.1
**AGE**	**N**	**%**
18–35	304	25.3
36–45	182	15.2
46–55	248	20.7
56–65	188	15.7
≥66	278	23.2

**Table 2 nursrep-15-00250-t002:** Frequency and type of visit.

**HAVE YOU VISITED YOUR PRIMARY HEALTH CARE CENTER IN THE LAST YEAR?**	**N**	**%**
Several times a month	144	12
Once a month	85	7.1
Several times a year	406	33.8
Once or twice a year	448	37.3
I haven’t been in the last one/two years	117	9.8
**APART FROM YOUR OWN NEEDS, DO YOU ACCOMPANY OTHER MEMBERS OF YOUR FAMILY TO THE PRIMARY HEALTH CARE CENTER?**	**N**	**%**
Yes	746	62.2
No	454	37.8
**(*) WHICH FAMILY MEMBER(S) DO YOU ACCOMPANY?**	**N**	**%**
Minor	325	43.5
Spouse/partner	312	41.8
Other dependent family members: elderly, disabled… etc.	373	50

* Multiple-response question. Base: those who answered “yes” to accompanying others.

**Table 3 nursrep-15-00250-t003:** Access to and experience of the nurse at the Primary Health Care Center.

**DO YOU KNOW IF YOU HAVE A DESIGNATED NURSE AT YOUR PRIMARY HEALTH CARE CENTER, JUST AS YOU HAVE A DESIGNATED PHYSICIAN?**	**N**	**%**
Yes	833	69.4
No	367	30.6
**HOW OFTEN DO YOU USUALLY VISIT THE NURSE AT YOUR PRIMARY HEALTH CARE CENTER, EITHER FOR YOURSELF OR FOR A FAMILY MEMBER?**	**N**	**%**
Once or more a month	162	13.5
Two or three times a year	366	30.5
Once a year	220	18.3
I haven’t been for more than a year	164	13.7
I have never had an appointment with the nurse	288	24
**(*) YOU USUALLY REQUEST YOUR APPOINTMENT WITH THE NURSE AT THE PRIMARY HEALTH CARE CENTER...**	**N**	**%**
By phone	348	38.1
Through an app or online	419	45.9
By going to the primary health care center in person	484	53
I visit the physician and he/she refers me to the nurse	418	45.8
**(**) ON OVERAGE, WHEN YOU REQUEST AN APPOINTMENT WITH THE NURSE AT THE PRIMARY HEALTH CARE CENTER, HOW MANY DAYS DOES IT TAKE FOR YOU TO BE SEEN?**	**N**	**%**
I am seen on the same day	153	16.7
1 day	115	12.6
2 days	191	20.9
More than 2 days	453	49.6
**(**) IN GENERAL, DO YOU THINK THE TIME YOUR NURSE SPENDS WITH YOU IS APPROPIATE?**	**N**	**%**
Adequate	815	89.3
I would need more time	97	10.6
**(*) IN ADDITION TO THE IN-PERSON CONSULTATION AT THE PRIMARY HEALTH CARE CENTER, HOW HAVE YOU HAVE YOU OR A MEMBER OF YOUR FAMILY RECEIVED CARE FROM NURSES AT ANY TIME? (YOU MAY SELECT MORE THAN ONE OPTION).**	**N**	**%**
At home	204	22.3
Over the telephone	365	40
In schools	76	8.3
In nursing homes	42	4.6
**(*) OF THE SERVICES OR TYPES OF CARE I AM GOING TO LIST, WHICH DO YOU THINK YOU OR A MEMBER OF YOUR FAMILY HACE RECEIVED FROM NURSES AT PRIMARY HEALTH CARE CENTERS IN THE LAST YEAR?**	**N**	**%**
Monitoring and tests: acenocoumarol monitoring, blood tests, etc.	649	71.1
Administration of treatments: injections, aerosol therapies, etc.	480	52.6
Wound care and suturing	380	41.6
Follow-up of chronic diseases or routine check-ups	397	43.5
Health education and disease prevention, either at the health center or in schools and nursing homes	137	15
Prescription of medical supplies (nappies, glucose test strips for diabetics, dressings, etc.)	203	22.2
Assistance with the management of complex administrative procedures: health insurance requests or medical reports	143	15.6
Requesting diagnostic tests	227	24.8
**HAVE YOU PARTICIPATED IN ANY GROUP ORGANISED BY YOUR PRIMARY HEALTH CARE CENTER TO PROVIDE INFORMATION ABOUT HEALTH CARE (SUCH AS DIABETES CARE, HYPERTENSION CARE, PREGNANCY, ETC.)**	**N**	**%**
(***) Yes	149	12.4
No	1051	87.6
**GROUP TOPIC**	**N**	**%**
Pregnancy and childbirth	105	70.4
Diabetes	25	16.7
Hypertension	7	4.7
Other topics	26	17.4

(*) Multiple-choice questions. Values may exceed 100%. Base: “Has visited the nurse’s consultation.” (**) Base: “Has visited the nurse’s consultation.” (***) Multiple-response question. Base: “Has participated.”.

**Table 4 nursrep-15-00250-t004:** A new model of patient–nurse relationship in Primary Health Care Centers.

**WOULD YOU LIKE THE SAME NURSE TO ALWAYS PROVIDE ALL THE CARE YOU NEED (SUCH AS PERFORMING TESTS, ADMINISTERING VACCINES, FOLLOWING-UP CARE OF CHRONIC CONDITIONS, ETC.)**	**N**	**%**
A lot/Quite a lot	1039	86.6
Very little/Not at all	161	13.4
**WOULD YOU LIKE THE SAME NURSE TO CARE FOR ALL THE MEMBERS OF YOUR FAMILY?**	**N**	**%**
A lot/Quite a lot	880	73.3
Very little/Not at all	320	26.7
**RECENTLY, THERE HAS BEEN A DISCUSSION IN THE MEDIA ABOUT NURSES BEING THE FIRST TO RECEIVE PATIENTS AT PRIMARY HEALTH CARE CENTERS—HAS BEEN THE CASE FOR YEARS FOR CERTAIN HOSPITAL EMERGENCIES—TO DEAL WITH SOME PROBLEMS OR REFER PATIENTS TO A PHYSICIAN IF NECESSARY. WOULD YOU...**	**N**	**%**
Agree with this procedure	399	33.3
Disagree	183	15.3
Depending on the case	618	51.5
**TO WHAT EXTENT DO YOU THINK IT IS APPROPRIATE FOR PATIENTS WITH EXPERIENCE AND TRAINING IN CERTAIN DISEASES OR HEALTH PROBLEMS TO COLLABORATE WITH NURSES TO HELP OTHER PATIENTS?**	**N**	**%**
Very appropriate	346	28.8
Fairly appropriate	448	37.3
Somewhat appropriate	270	22.5
Not at all appropriate	136	11.3

**Table 5 nursrep-15-00250-t005:** Knowledge and acceptance of new competencies.

**DID YOU KNOW THAT NURSES IN PRIMARY HEALTH CARE CENTERS CAN PRESCRIBE PRODUCTS SUCH AS PARACETAMOL, IBUPROFEN, OMEPRAZOLE, HYDROCORTISONE, ANTIDIABETICS, ANTICOAGULANTS, …?**	**N**	**%**
Yes	463	38.6
No	594	49.5
Not sure	143	11.9
**DO YOU THINK IT IS NECESSARY FOR THE NURSE AT THE PRIMARY HEALTH CARE CENTER TO BE ABLE TO ORDER DIAGNOSTIC TESTS FOR THE FOLLOW-UP AND MANAGEMENT OF CHRONIC DISEASES (SUCH AS HYPERTENSION, DIABETES** **,** **...)?**	**N**	**%**
Yes	840	70
No	129	10.8
Not sure	231	19.3
**DO YOU THINK IT IS NECESSARY FOR THE NURSE AT THE PRIMARY HEALTH CARE CENTER TO BE ABLE TO PRESCRIBE THE NECESSARY MEDICATION FOR THE FOLLOW-UP AND MANAGEMENT OF CHRONIC DISEASES (SUCH AS HYPERTENSION, DIABETES, …?**	**N**	**%**
Yes	766	63.8
No	217	18.1
Not sure	217	18.1
**DO YOU AGREE THAT NURSES SHOULD RECEIVE PATIENTS IN PRIMARY HEALTH CARE CENTERS, TAKE CARE OF THEM AND REFER THEM TO A PHYSICIAN IF NECESSARY?**	**N**	**%**
Agree	408	34.0
Depending on the case	612	51.0
Disagree	180	15.0
**DO YOU THINK IT IS NECESSARY FOR THE NURSE AT THE PRIMARY HEALTH CARE CENTER TO CONTACT PATIENTS OR THEIR FAMILY MEMBERS AFTER A HOSPITAL DISCHARGE?**	**N**	**%**
Yes	1024	85.3
No	67	5.6
Not sure	109	9.1
**WOULD YOU LIKE THE NURSE WHO ATTENDS TO YOU AT THE PRIMARY HEALTH CARE CENTER TO BE A SPECIALIST NURSE WITH SPECIFIC TRAINING AND COMPETENCIES IN PROVIDING HEALTH CARE TO INDIVIDUALS, FAMILIES, AND THE COMMUNITY AT ALL STAGES OF LIFE?**	**N**	**%**
Yes	988	82.3
No	18	1.5
I have no preference	194	16.2
**DO YOU THINK THAT A NURSE WITH MANAGEMENT QUALIFICATIONS COULD LEAD THE PRIMARY HEALTH CARE CENTER?**	**N**	**%**
Yes	736	61.3
No	253	21.1
Not sure	211	17.6

**Table 6 nursrep-15-00250-t006:** Analysis of variables by gender and age.

	**Female**		**Male**		***p* ^T^**	**18–35**		**36–45**		**46–55**		**56–65**		**≥66**		***p* ^A^**
	N	%	N	%		N	%	N	%	N	%	N	%	N	%	
**DO YOU KNOW IF YOU HAVE A DESIGNATED A NURSE AT YOUR PRIMARY HEALTH CARE CENTER, JUST AS YOU HAVE A PRIMARY HEALTH CARE PHYSICIAN?**																
Yes	532	73.3	300	63.4	0.001 *	185	60.9	133	73.1	178	71.8	144	76.6	193	69.4	0.002 *
No	194	26.7	173	36.6		119	39.1	49	26.9	70	28.2	44	23.4	85	30.6	
**HOW OFTEN DO YOU USUALLY VISIT THE NURSE AT YOUR PRIMARY HEALTH CARE CENTER, EITHER FOR YOURSELF OR FOR A FAMILY MEMBER?**																
One or more times per month	111	15.3	51	10.8	0.238	27	8.9	23	12.6	42	16.9	32	17	38	13.7	<0.001 *
Two or three times a year	212	29.2	153	32.3		71	23.4	58	31.9	65	26.2	64	34	108	38.8	
Once a year	131	18	89	18.8		53	17.4	27	14.8	49	19.8	35	18.6	56	20.1	
I have not visited for over a year	107	14.7	57	12.1		57	18.8	26	14.3	34	13.7	22	11.7	25	9	
I have never made an appointment with the nurse	165	22.7	123	26		96	31.6	48	26.4	58	23.4	35	18.6	51	18.3	
**IN GENERAL, DO YOU THINK THE TIME YOUR NURSE SPENDS WITH YOU IS APPROPIATE?**																
Adequate	491	87.5	324	92.6	0.001 *	188	90.4	113	84.3	160	84.2	138	90.2	216	95.2	0.002 *
I would like more time	70	12.5	26	7.4		20	9.6	21	15.7	30	15.8	15	9.8	11	4.8	
**HAVE YOU PARTICIPATED IN ANY GROUP ORGANIZED BY YOUR PRIMARY HEALTH CARE CENTER TO PROVIDE INFORMATION ABOUT HEALTH CARE (SUCH AS DIABETES CARE, HYPERTENSION MANAGEMENT CARE, PREGNANCY, ETC.)?**																
Yes	119	16.4	30	6.3	<0.001 *	33	10.9	48	26.4	41	16.5	15	8	12	4.3	<0.001 *
No	607	83.6	443	93.7		271	89.1	134	73.6	207	83.5	173	92	266	95.7	
**WOULD YOU LIKE THE SAME NURSE TO ALWAYS PROVIDE ALL THE CARE YOU NEED (SUCH AS PERFORMING TESTS, ADMINISTERING VACCINES, FOLLOWING-UP CARE FOR CHRONIC CONDITIONS, ETC.)?**																
A lot/Quite a lot	186	67.9	88	32.1	0.581	102	72.3	45	60.8	62	65.3	52	72.2	85	70.8	0.379
Very little/Not at all	88	32.1	68	29.8		39	27.7	29	39.2	33	34.7	20	27.8	35	29.2	
**WOULD YOU LIKE THE SAME NURSE TO CARE FOR ALL MEMBERS OF YOUR FAMILY?**																
A lot/Quite a lot	211	54	180	46	0.843	113	56.8	49	41.9	75	55.1	51	56	73	56.2	0.093
Very little/Not at all	150	53.2	132	46.8		86	43.2	68	58.1	61	44.9	40	44	57	43.8	
**RECENTLY, THERE HAS BEEN A DISCUSSION IN THE MEDIA ABOUT NURSES BEING THE FIRST TO RECEIVE PATIENTS AT PRIMARY HEALTH CARE CENTERS—HAS BEEN THE CASE FOR YEARS FOR CERTAIN HOSPITAL EMERGENCIES—TO DEAL WITH SOME PROBLEMS OR REFER PATIENTS TO A PHYSICIAN IF NECESSARY. WOULD YOU…**																
In agreement with this procedure	212	29.2	186	39.3	0.004 *	105	34.5	52	28.6	67	27	70	37.2	105	37.8	0.004 *
In disagreement	118	16.3	65	13.7		30	9.9	31	17	41	16.5	28	14.9	53	19.1	
Depending on the case	396	54.5	222	46.9		169	55.6	99	54.4	140	56.5	90	47.9	120	43.2	
**WOULD YOU LIKE THE NURSE WHO ATTENDS TO YOU AT THE PRIMARY HEALTH CARE CENTER TO BE A SPECIALIST NURSE WITH SPECIFIC TRAINING AND COMPETENCIES IN PROVIDING HEALTH CARE TO INDIVIDUALS, FAMILIES AND THE COMMUNITY AT ALL STAGES OF LIFE?**																
Yes	604	83.2	383	81	0.626	242	79.6	151	83	217	87.5	157	83.5	221	79.5	0.145
No	8	1.1	10	2.1		3	1	4	2.2	3	1.2	1	0.5	7	2.5	
I have no preference.	114	15.7	80	16.9		59	19.4	27	14.8	28	11.3	30	16	50	18	
**DO YOU THINK A NURSE WITH MANAGEMENT QUALIFICATIONS COULD LEAD THE PRIMARY HEALTH CARE CENTER?**																
Yes	463	63.8	272	57.5	0.143	213	70.1	118	64.8	159	64.1	119	63.3	127	45.7	<0.001 *
No	137	18.9	116	24.5		44	14.5	34	18.7	52	21	38	20.2	85	30.6	
I am not sure	126	17.4	85	18		47	15.5	30	16.5	37	14.9	31	16.5	66	23.7	
**HOW APPROPRIATE DO YOU THINK IT IS FOR PATIENTS WITH EXPERIENCE AND TRAINING IN CERTAIN DISEASES OR HEALTH PROBLEMS TO WORK WITH NURSES TO HELP OTHER PATIENTS?**																
Very appropiate/Quite appropiate	490	67.5	304	64.3	0.193	205	67.4	128	70.3	173	69.8	123	65.4	165	59.4	0.064
Somewhat appropiate/Not appropiate at all	236	32.5	169	35.7		99	32.6	54	29.7	75	30.2	65	34.6	113	40.6	

^T^: *t*-test; ^A^: ANOVA test; * significance level *p* < 0.05.

**Table 7 nursrep-15-00250-t007:** Multiple logistic regressions of factors associated with knowledge and perception of the nursing role in Primary Health Care.

	OR	IC95%		B	Wald	*p* *	R ^1^ Nagelkerke
**IS AWARE OF HAVING A DESIGNATED REFERENCE NURSE**							
Age 18–35 years	0.595	0.472	0.751	0.665	14.469	<0.001 **	0.043
Age 56–65 years	1.677	1.129	2.492	−0.366	2.335	0.127	
Public healthcare	1.110	1.001	1.232	−0.358	4.658	0.031 **	
**PARTICIPATION IN WORKING GROUPS AT YOUR PRIMARY HEALTH CARE CENTER**							
Higher education	1.316	1.091	1.587	−0.354	2.500	0.114	0.132
Age 36–45 years	2.874	2.080	3.972	−1.133	21.326	<0.001 **	
Age >66 years	0.341	0.173	0.674	0.715	3.260	0.071	
Knows designated reference nurse	1.183	1.064	1.315	−0.618	5.261	0.022 **	
Male	1.340	1.192	1.506	−0.769	8.772	0.003 **	
Contact with healthcare professionals	1.255	1.034	1.523	−0.337	2.293	0.130 **	
**HAVING THE SAME NURSE FOR ALL YOUR CARE**							
Age 36–45 years	0.539	0.312	0.929	0.925	7.183	0.007 **	0.101
Knows designated reference nurse	1.506	1.207	1.878	−1.111	19.008	<0.001 **	
**HAVING THE SAME NURSE FOR ALL FAMILY MEMBERS CARE**							
Age 36–45 years	0.496	0.328	0.750	0.878	11.467	0.001 **	0.072
Public healthcare	1.196	1.058	1.352	−0.489	5.466	0.019 **	
Knows designated reference nurse	1.162	1.020	1.325	−0.528	6.047	0.014 **	
**NURSE WITH MANAGEMENT TRAINING TO LEAD A PRIMARY HEALTH CARE CENTER**							
Age 18–35 years	1.607	1.163	2.219	0.506	2.298	0.130	0.07
Age >66 years	0.520	0.399	0.677	−0.483	4.784	0.029 **	
Elementary education	0.470	0.262	0.843	0.65	10.644	0.001 **	
Female	0.806	0.672	0.967	0.278	2.688	0.101	
Knows designated reference nurse	1.169	1.040	1.314	−0.547	9.689	0.002 **	

^1^: R square of Nagelkerke. * *p*: significance level of the variable within the regression model. ** significant variables in the model. Note: categorical variables were coded using a one-vs-rest dummy approach; each OR compares the specified category with all other categories of the same variable combined.

## Data Availability

The raw data supporting the conclusions of this article will be made available by the authors on request.

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
