# Peer review of "A New Primary Care Model Based on Population Needs: A Nationwide Cross-Sectional Study"

_nursrep, 2025, doi:10.3390/nursrep15070250_

Round 1

Reviewer 1 Report

Comments and Suggestions for Authors

The manuscript presents a very clear and pertinent proposal.

The content is presented and systematised in a well-supported way, and the presentation of the objectives, as well as the development of the argument, is developed in well-articulated and coherent manner.

The overall quality of the analysis is positive, but it is possible to point out some aspects that could be improved.

The introductory section would benefit from some historical, political, and professional contextualisation of the emergence, development, and recent transformations of Primary Health Care in Spain. Not only does this help to situate the reader who is unfamiliar with the Spanish reality, but it also makes it possible to pinpoint more precisely the scope and deeper impact that the 2005 reform has had on Primary Health Care. In other words, what has changed?

One aspect that is omitted throughout the text is the issue of interprofessional relations between nursing and medicine. Once again, for a reader unfamiliar with the Spanish reality, the question of the particularities of this context remains, since the evolution of primary care nursing seems to be the result of growth that presupposes a new balance in the division of labour and the sharing of professional power. Some of nursing's new competences and responsibilities appear to be asserting themselves in areas whose jurisdiction (for example, prescribing) has traditionally been considered medical. Is this coexistence without tensions? Does it correspond to a fully stabilised logic of sharing, or with elements of some jurisdictional dispute?

Regarding the recruitment of the participants in the sample, it's not entirely clear how this happened. Why by telephone and how did it happen in operational terms? In other words, how was access gained to the participants' telephone contacts and how did this not violate any principles of the GDPR?

What is argued between lines 346 and 348 could be better discussed and substantiated, as it is an idea that could be seen as potentially debatable or at least open up some more problematising reflection. This professional expansion in the field of disease management could lead to an excessively vigilant profile and, consequently, a medicalised approach to citizens' health.

Finally, there are just two formal details in tables 3 and 7. In table 3 it looks like ‘si’ instead of “yes” and in table 7, “años” instead of ‘years’.

Reviewer 2 Report

Comments and Suggestions for Authors

This study explores the perceptions and expectations of adult primary care users in Spain regarding the role of nurse specialists in primary health care, using data from a nationwide telephone survey. A key strength of the study is its large, randomly selected sample, which enhances the generalizability of the findings. However, the manuscript would benefit from improved clarity in methodological reporting, more consistent presentation of statistical results, and clearer integration of findings into a broader primary care framework.

  • Language consistency: Some Spanish terms remain untranslated. For example, in line 184, “Abril” should be “April,” and in Table 3 (the first line in page 9), the word “si” should be “yes.” Please review the manuscript to ensure consistency in language throughout.
  • Sample size calculation: In lines 184–185, it is stated that a sample size of 385 surveys was calculated based on a 95% confidence level and a 5% margin of error. Please provide more detail on how this calculation was performed. Was the sample size based on the total resident population of Spain (N = 48,692,804) or on a population that met specific inclusion criteria? If based on the total population, please clarify how weighting was handled given that the actual sample likely represents only a subset. If based on the eligible population, please report how many individuals met the inclusion criteria.
  • Inclusion/exclusion criteria: The criteria listed in lines 187–190 are unclear. It is not specified whether individuals over 18 years old and residing in Spain who declined to participate are considered part of the study population that the sample is intended to represent. If they are included in the denominator for representativeness, it would be useful to compare their demographic characteristics with those who agreed to participate to assess potential selection bias. If not, please report the percentage of eligible individuals who declined to participate.
  • Incomplete sentence (line 260–261): The sentence beginning “knowledge of the designated nurse is…” appears incomplete and lacks a comparison group. Please revise for clarity.
  • Clarification needed (line 263): The phrase “of those who have visited the nurse at least once” is vague. Please clarify whether this refers to at least once per month or per year.
  • Appointment wait times (Table 3): For the question on average wait time for a nurse appointment, nearly half of respondents reported waiting more than two days. Were specific numbers of days collected? If so, it would be informative to report the median or range of wait times.
  • Footnote (Table 3): Table 3 includes a footnote marked with “***” that does not appear to correspond to any content in the table. Please revise the table to include the appropriate reference for this footnote or remove it if it is not needed.
  • Table 7 (Regression models): The title of Table 7 refers to “multivariate analysis,” but the models presented appear to be multiple logistic regressions, as all outcome variables are binary. A true multivariate logistic regression would involve outcome variables with more than two categories. Please revise the title accordingly, unless any of the models include non-binary outcomes. Additionally, the reference groups for categorical variables are not clearly indicated. For example, it is unclear which age group is used as the reference category. It also appears that different models may use different reference groups for the same variable, which could be confusing. For clarity and consistency, please use the same reference groups across models and clearly specify them in the table or footnotes.
  • Model visualization: Since the manuscript aims to support or inform a new primary care model, it would be helpful to include a visual representation of the proposed model. This could highlight the various components and how they function together within the system, particularly the role of the nurse specialist in care coordination and service delivery.
  • Discussion improvement: While the discussion section addresses individual aspects of the nurse specialist’s role, it would be strengthened by integrating findings into a broader systems-level perspective. Consider connecting the results to existing health systems or care delivery frameworks to better demonstrate how the proposed model can enhance primary care outcomes and patient experience.

Reviewer 3 Report

Comments and Suggestions for Authors

Dear Authors,

Thank you for the opportunity to review your manuscript "NEW PRIMARY CARE MODEL: POPULATION NEEDS. A NATIONWIDE CROSS-SECTIONAL STUDY." This study addresses an important topic regarding public perceptions of expanded nursing roles in primary care settings. While the large-scale national survey provides valuable insights, I would like to offer the following comments for your consideration:

Major Comments:

  1. Conceptual clarity: The inconsistent use of "primary care" and "primary health care" throughout the manuscript requires clarification. These terms have distinct meanings as established by the Alma-Ata Declaration, and consistent terminology would strengthen the theoretical framework.
  2. Comparative perspective: The study would benefit significantly from including comparative questions between physicians and nurses. Without data on physician recognition rates (likely >95%) compared to nurse recognition (69.4%), the relative "invisibility" of nurses cannot be properly contextualized. More critically, direct preference comparisons for specific services (prescribing, diagnostic tests, family care) would provide essential policy-relevant information.
  3. Statistical concerns: Table 7 appears to show R values rather than R² values as would be expected for logistic regression. Regardless, the very low values (0.043-0.132) suggest that important predictors are missing from the models. Variables such as personal health status, previous healthcare experiences, media exposure, and regional healthcare infrastructure should be considered in future analyses.
  4. Transparency issues: The response rate is not reported, which is essential for assessing potential non-response bias. Additionally, the female predominance (60.5%) warrants discussion regarding representativeness.

Minor Comments:

  1. International accessibility: While I understand the Spanish context, incorporating more English-language references alongside Spanish sources would enhance the manuscript's international relevance and allow for better evaluation of the literature review.
  2. Discussion gaps: The remarkably low model explanatory power deserves thorough discussion. Additionally, the gender differences in bivariate versus multivariate analyses (Simpson's paradox) and the high opposition rate among elderly respondents (45.7%) require more nuanced interpretation.
  3. Policy implications: Without comparing preferences for "increasing primary care physicians" versus "expanding nurse roles," the study cannot adequately inform resource allocation decisions in healthcare policy.

Recommendations:

Despite these limitations, the study makes a valuable contribution by documenting public acceptance of expanded nursing roles.

I recommend revisions to address the comparative perspective and statistical concerns. Future research should directly compare physician and nurse preferences for various services and include policy priority rankings to better inform healthcare system reforms.

The manuscript demonstrates considerable effort in surveying a large national sample and addresses a timely issue. With the suggested revisions, it could provide more robust evidence for policy decisions regarding primary care workforce development.

Sincerely,

A reviewer

Round 2

Reviewer 2 Report

Comments and Suggestions for Authors

Thank you for the thoughtful revisions. The discussion section has been greatly improved and provides a clearer interpretation of the study’s implications. However, I still have several comments regarding the Methods and Results sections, particularly related to the reporting and interpretation of data analysis. These clarifications are important to enhance the overall transparency and rigor of the manuscript.

  1. Clarification of Inclusion and Participation Numbers (Lines 280–281):

The current phrasing is unclear. The manuscript states:

“A total of 40,000 calls were made, of which 0.7% (280) met the inclusion criteria and didn’t accept to participate. A total of 1,200 individuals participated in the study.”

If the total number of eligible individuals was 1,480 (1,200 participants + 280 non-participants), it would be more accurate to report the refusal rate based on eligible respondents (280/1,480 = 18.9%) rather than total calls. I suggest revising the sentence to something like:

“Of the 40,000 calls made, 1,480 individuals met the inclusion criteria, of which 280 (18.9%) declined to participate.”

This would improve transparency around the recruitment process and participation rate.

  1. Appointment Wait Time – Table 3:

Thank you for including the median and interquartile range for appointment wait times. However, the reported values appear inconsistent with the distribution shown in Table 3. Since over 50% of respondents waited 2 days or less, the median should logically be ≤2 days. The reported median of 7 days with an IQR of 3.5 seems incorrect. Please double-check the analysis and ensure the reported summary statistics accurately reflect the data.

  1. Table 7 – Multiple Logistic Regression:

While the correction of the table title is appreciated, there are still important issues to address:

  • Reference Groups Missing: The table lacks identification of the reference groups for categorical variables, which is essential for interpreting odds ratios. For example, in Lines 358–360, the authors note that “awareness is significantly more frequent among individuals aged 56–65 years,” but it is unclear which age group serves as the reference. Please include reference categories in both the table and the interpretation of results to ensure clarity.
  • Clarify Column ‘R’: The final column contains an “R” with values that resemble statistical metrics. If this refers to R-squared or another model fit statistic, please clarify its meaning in the table footnotes or main text. If not, consider relabeling or removing this column to avoid confusion.

I recommend consulting a biostatistician or data analyst to ensure the correct construction and interpretation of Table 7. These changes will significantly strengthen the manuscript’s methodological transparency and credibility.

Reviewer 3 Report

Comments and Suggestions for Authors

The authors have made substantial improvements to the manuscript through extensive revisions. Most of the raised concerns have been adequately addressed, significantly enhancing the quality of the research.

However, I noticed two issues that require attention:

  1. Figure 1 citation: Figure 1 appears to be presented without proper citation in the main text. All figures must be referenced and discussed within the manuscript body according to journal guidelines. Please ensure Figure 1 is appropriately cited and described in the relevant section.
  2. Inconsistent terminology: While the authors state that "PC" has been changed to "PHC," several instances of "PC" remain in the Discussion section. Please conduct a thorough search and replace all remaining instances of "PC" with "PHC" to maintain consistency throughout the manuscript.

These are relatively minor but important issues that, once corrected, will result in a well-polished manuscript ready for publication.
